# Protective Effect of Avenanthramide-C on Auditory Hair Cells against Oxidative Stress, Inflammatory Cytokines, and DNA Damage in Cisplatin-Induced Ototoxicity

**DOI:** 10.3390/ijms24032947

**Published:** 2023-02-02

**Authors:** Alphonse Umugire, Yoon Seok Nam, Ye Eun Nam, Young Mi Choi, Se Myeong Choi, Sungsu Lee, Jong Hyun Cho, Hyong-Ho Cho

**Affiliations:** 1Department of Otolaryngology-Head and Neck Surgery, Chonnam National University Hospital, Chonnam National University Medical School, Gwangju 61469, Republic of Korea; 2Department of Biomedical Science, College of Medicine, Chonnam National University Graduate School, Gwangju 61469, Republic of Korea; 3BK21 PLUS Center for Creative Biomedical Scientists at Chonnam National University, Gwangju 61469, Republic of Korea; 4Department of Medicinal Biotechnology, College of Health Sciences, Dong-A University, Busan 49315, Republic of Korea; 5Department of Translational Biomedical Sciences, Graduate School of Dong-A University, Busan 49201, Republic of Korea

**Keywords:** avenanthramide-C, cisplatin, DNA damage, ROS, hair cell loss, hearing loss

## Abstract

Cisplatin-induced ototoxicity leads to hearing impairment, possibly through reactive oxygen species (ROS) production and DNA damage in cochlear hair cells (HC), although the exact mechanism is unknown. Avenanthramide-C (AVN-C), a natural, potent antioxidant, was evaluated in three study groups of normal adult C57Bl/6 mice (control, cisplatin, and AVN-C+cisplatin) for the prevention of cisplatin-induced hearing loss. Auditory brainstem responses and immunohistochemistry of outer hair cells (OHCs) were ascertained. Cell survival, ROS production, Phospho-H2AX-enabled tracking of DNA damage-repair kinetics, and expression levels of inflammatory cytokines (*TNF-α*, *IL-1β*, *IL6*, *iNOS*, and *COX2*) were assessed using House Ear Institute-Organ of Corti 1 (HEI-OC1 Cells). In the in vivo mouse model, following cisplatin-induced damage, AVN-C decreased the hearing thresholds and sheltered all cochlear turns’ OHCs. In HEI-OC1 cells, AVN-C preserved cell viability and decreased ROS production, whereas cisplatin enhanced both ROS levels and cell viability. In HEI-OC1 cells, AVN-C downregulated *IL6*, *IL-1β*, *TNF-α*, *iNOS*, and *COX2* production that was upregulated by cisplatin treatment. AVN-C attenuated the cisplatin-enhanced nuclear H2AX activation. AVN-C had a strong protective effect against cisplatin-induced ototoxicity through inhibition of ROS and inflammatory cytokine production and DNA damage and is thus a promising candidate for preventing cisplatin-induced sensorineural hearing loss.

## 1. Introduction

Cisplatin is an effective anticancer treatment, although the ototoxic potential of cisplatin therapy exposes cancer patients to the risk of hearing loss, which can further impair the patient’s quality of life [1]. Since its introduction, cisplatin has been one of the most potent cancer chemotherapeutic agents in both children and adults. Cisplatin remains unique and has unsurpassed efficacy against a broad range of tumors (including osteogenic sarcoma, medulloblastoma, testicular, cervical, and ovarian cancers) [2]. Moreover, cisplatin has a broad toxicity profile that affects the gastrointestinal, hematologic, renal, and auditory systems [3], and induces DNA damage [4].

Such DNA damage leads to the production of single-stranded DNA (ssDNA) and double-stranded DNA (dsDNA), and these are repaired by gamma H2AX, a phosphorylated histone H2AX which is a marker for DNA repair that triggers histone H2AX phosphorylation in broad regions of chromatin that flank the strand breaks [5]. Phosphorylation, along with acetylation-dependent chromatin condensation, enables microscopic imaging of distinct nuclear foci [6]. Thus, the creation of phosphorylated H2AX foci is a potent tool for studying double-strand break (DSB) formation and repair after genomic damage, as well as chromosomal dynamics and signaling processes [7,8].

Avenanthramides (AVNs) are low-molecular-weight phenolic compounds that are derived from oat grain (*Avena sativa* L.) and have strong antioxidant properties, which are often found in human food. Avenanthramide-C (AVN-C) has the maximum antioxidant activity [9] and protects against both noise-induced hearing loss (NIHL) and drug-induced hearing loss (DIHL) [10]. In mice, AVN-C attenuated methotrexate-induced hearing loss [11].

This study was conducted with the aim to evaluate the ability of AVN-C to inhibit the harmful effect of reactive oxygen species (ROS) and reduce the inflammation caused by cisplatin, as well as to determine whether AVN-C is involved in DNA repair over time, as this would help determine the potential applicability of AVN-C as a novel therapy for preventing cisplatin-induced hearing loss.

## 2. Results

### 2.1. AVN-C Protects against Hearing Loss due to Cisplatin-Induced Ototoxicity

Hearing sensitivity was significantly altered by cisplatin treatment. Figure 1B,C demonstrate that the cisplatin-only treated group had higher auditory brainstem response (ABR) threshold shifts than the controls, which is an indirect indicator of OHC function. The ABRs were measured to determine auditory sensitivity. Mice hearing was evaluated prior to injecting AVN-C and/or cisplatin, as well as at the end of the treatment protocol. Figure 1B depicts the elevated threshold shifts (46 ± 5.5 dB SPL) in the cisplatin group and in the Control (20 ± 0 dB SPL), whereas AVN-C decreased the ABR threshold shifts (27 ± 4.4 dB SPL) for the click sound. Control animals (5% Kalliphor-injected) showed no significant differences in hearing sensitivity at tone-burst frequencies (24 ± 2.2, 23 ± 2.7, 28 ± 2.7, and 28 ± 2.7 dB SPL at 8, 16, 24, and 32 kHz, respectively; *p* ≤ 0.001; Figure 1C). However, mice that were administered 5 mg/kg cisplatin four times consecutively (Figure 1A) displayed substantially increased threshold shifts across the whole tone-burst frequency ranges examined, and the threshold shifts were more severely affected at 24 and 32 kHz (80 ± 9.4 and 92 ± 11 dB SPL, respectively; Figure 1C). The ABR results of the mice that received AVN-C and cisplatin were significantly lower in threshold shift values (28 + 2.7, 27 + 2.7, 39 + 2.2, and 49 + 4.2 dB SPL at 8, 16, 24, and 32 kHz, respectively; Figure 1C). ABR at 1 and 2 weeks were checked to determine cisplatin-induced hearing loss and the data are shown in Appendix A.

### 2.2. AVN-C Prevents Cisplatin-Induced OHC Depletion

Following the last hearing test, mice were euthanized and their cochleae were excised and examined. Figure 2A illustrates representative maximum-intensity projections of confocal stacks of the organ of Corti of the three study groups (control, CP, and AVN-C+CP). Inner hair cells (IHCs) were distinguished by their Calbindin-labeled cell bodies (green), whereas OHCs were tagged with Prestin (red). OHCs in cisplatin-treated mice displayed a significant decrease at the base region, whereas the mid- and apical-cochlear regions had a moderate loss of OHCs compared to the control group (*** *p* ≤ 0.001, control vs. cisplatin-treated group).

The AVN-C pretreatment before cisplatin administration significantly enabled the retention of a large number of OHCs in the whole cochlear region (apex, middle, and base regions; *** *p* ≤ 0.001, AVN-C vs. cisplatin treatments). In control mice, the OHCs remained unharmed. The IHCs appeared normal and were well maintained in all groups (Figure 2B).

### 2.3. AVN-C Prevents Cisplatin-Induced Synaptic Ribbon Degeneration

Immediately after the ABR readings, the synaptic ribbon was examined in cochlear whole-mount specimens and the cochleae were stained with RIBEYE/CtBP2 and Myosin7a; next, the presynaptic ribbons were quantified to estimate the extent of CP damage and the potential of AVN-C to rescue these presynaptic ribbons. Thus, we investigated the regions corresponding to the tested stimulus frequencies that evoked ABR and measured the presynaptic ribbons. CP considerably decreased the number of IHC presynaptic ribbons throughout the cochlea, and this was particularly prominent at the base turn (Figure 3A). The corresponding CtBP2-positive signal counts were observed: apex turn (6 ± 0.6), middle turn (4 ± 1), and base turn (3 ± 0.6) presynaptic ribbons per 20 µm IHC (Figure 3B). AVN-C pretreatment before CP injection conferred a better protective impact on hearing thresholds, and a greater number of presynaptic ribbons was saved from CP-induced damage in the following order (Figure 3B): apex turn (10 ± 1), middle turn (10 ± 0.6), and base turn (11 ± 2). The presynaptic ribbons were determined as 13 ± 1, 12 ± 1.5, and 12 ± 0.6 at the apex, middle, and base turns, respectively, in the control group that received 5% Kolliphor (AVN-C carrier; Figure 3B).

### 2.4. Trend in the Dose-Dependent Cytotoxicity of AVN-C and Cisplatin on HEI-OC1 Cells

After all treatments had ended, the survival of HEI-OC1 cells was assessed using a concentration-dependent technique (MTT assay). We first measured the concentrations of AVN-C (Figure 4A) and CP (Figure 4B) in HEI-OC1 cells by assessing several dosing regimens before determining the optimal AVN-C (Appendix A) and CP doses for the in vitro investigations. Precisely 24 h after all treatment protocols and the MTT assay were completed, the absorbance of HEI-OC1 cells was measured at 570 nm and indicated that cell survival was 100 ± 0% in the control, 75 ± 4.8% in the AVN-C+CP-treated group, and significantly decreased (53.1 ± 4.5%) in the CP-treated HEI-OC1 cells (Figure 4C). Further examination of cell survival on the cancer cells SNU-1014 and FaDu using 3 µmol AVN-C and various CP doses revealed a reduction in cell viability that was unaffected by AVN-C in these cancer cell types (Figure 4D,E).

### 2.5. AVN-C Suppresses Cisplatin-Induced ROS Production

The fluorescein-DCFDA channel-positive population of HEI-OC1 cells that were treated with CP produced significantly more ROS than the control group (32.2 + 2.5 vs. 6 + 1.6, CP vs. control, respectively). Thus, treatment with AVN-C 1 h before CP greatly decreased ROS generation (13.4 + 1.6 vs. 32.2 + 2.5, AVN-C+CP vs. CP alone; Figure 5A,B).

### 2.6. AVN-C Alleviates Cisplatin-Induced Inflammation in HEI-OC1 Cells

Based on the alterations in the inflammatory cytokine expression patterns, qualitative real-time polymerase chain reaction (RT-PCR) was used to better understand the downstream signaling pathways of AVN-C and CP. When the experiment was repeated, CP increased the expression of all the inflammatory genes that were examined (*IL6*, *IL-1β*, *TNF-α*, *iNOS*, and *COX2*). AVN-C pretreatment before introducing CP to HEI-OC1 cells appeared to considerably reduce inflammation by decreasing the expression of these genes (*IL6*, *IL-1β*, *TNF-α*, *iNOS*, and *COX2*) (Figure 5C). The experiment was repeated three times and *p* ≤ 0.05 was found to be significant in all groups that were compared.

### 2.7. AVN-C Inhibits Cisplatin-Induced DNA Damage

Immunofluorescence microscopy was used to measure time-dependent phosphorylated H2AX levels related to AVN-C and CP exposure at various time points. At 24 h after exposure to HEI-OC1 cells, the intensity of H2AX fluorescence in AVN-C-exclusively treated cells was equal to that of non-treated cells, with a 1-fold increase in cells and a small number of low nuclear H2AX foci. Three hours after CP addition, cells displayed varied nuclear H2AX phosphorylation patterns that were clearly visible in the AVN-C-treated group, along with a 1.4-fold increase (Figure 6A,B). CP induced a rapid increase in H2AX levels within 12 h, whereas AVN-C significantly reduced H2AX foci-positive cells (*** *p* ≤ 0.001, AVN-C vs. CP-treated groups; Figure 6A,B).

### 2.8. Proposed AVN-C Mechanism of Action in the Prevention of Cisplatin-Induced Ototoxicity

CP induces the synthesis of pro-inflammatory cytokines such as tumor necrosis factor-alpha (*TNF-α*), interleukin-1 (*IL-1*), and nuclear factor kappa β after entering cochlea cells by passive diffusion and aided transport (NF-κB), and these substances are extremely toxic. The generation of ROS by NOX3, which stimulates the pathways for lipid peroxidation, DNA damage, inflammation, and cell death, is one of the factors that contribute to CP’s ototoxicity. This toxic event is blocked by AVN-C treatment through scavenging ROS (antioxidants), lowering inflammation, and repairing DNA damage (Figure 7).

## 3. Discussion

Oncologic medications have long been known as constituting ototoxic agents. Every year, more than 500,000 individuals worldwide are diagnosed with cancer and more than 100,000 die of cancer [12]. Platinum-based anticancer medications and aminoglycoside antibiotics are the most significant first-line cancer therapeutics, and these are potentially harmful to the inner ear [13]. In this regard, the relative ototoxicity of various platinum anticancer drugs, drug-dosing conditions that exacerbate ototoxicity, the cochlear structures that are most vulnerable to platinum damage, therapies for reducing ototoxicity, and the molecular signaling pathways that lead to cell death in the inner ear have all been studied extensively during the last 30 years [14,15,16,17]. Although all these agents have a platinum backbone, CP may have very different ototoxic effects. Some of the variances are most likely attributable to the chemical composition of the drugs, whereas individual patient features also play a role because an ototoxic pharmacological dose in one person may not be ototoxic in another [18,19]. 

Our findings revealed that CP induced irreversible hearing loss in mice at both click and tone-burst frequencies 1 month after intraperitoneal (IP) treatments (Figure 1B,C). Furthermore, a large loss of OHCs was observed, which was more prominent at the base, middle, and apex turns in B6 wild-type mice (Figure 2A,C); however, the IHCs were unharmed (Figure 2B). Second, AVN-C pretreatment before CP administration demonstrated significant hearing protection by lowering hearing thresholds to mimic the hearing results of the control group and thus prevented CP-induced ototoxicity. Notably, AVN-C secured OHCs and protected a greater number of OHCs against the harmful effects of CP (Figure 1B,C and Figure 2A,C).

Furthermore, CP is known to harm hair cells, with OHCs being more vulnerable than IHCs [20,21]. Thus, CP-induced hearing loss is typically bilateral and begins at high frequencies, before advancing to lower frequencies with sustained treatment [22,23]. Symptoms usually emerge days or weeks after beginning therapy and can persist long after treatment discontinuation [24]. Much research has focused on the prevention of DIHL in CP-induced hearing loss. In in vivo animal experiments, alpha-lipoic acid [25], dunnione7 [26], R-phenylisopropyladenosine [27], bucillamine [28], and forskolin [17] demonstrated protective effects against CP-induced ototoxicity, and AVN-C now joins this cluster. We previously measured and demonstrated that AVN-C penetrates the perilymph and passes the blood–labyrinth barrier (BLB) to protect mice from NIHL and DIHL; however, AVN-C-only treatment showed no effect on ABR and ROS production [10]. This may help us to understand why AVN-C did not show side effects and how it protects the inner ear organ.

Additionally, the synaptic ribbons that connect spiral ganglion neurons (SGNs) to IHCs are important synaptic elements in the sound conduction pathway and are crucial for the transmission of sound signals [29]. The loss of ribbon synapses affects how sound is transmitted and processed in the brain by raising hearing thresholds and resulting in hearing loss [30]. Furthermore, ribbon synapses can be damaged by various ototoxic medications. In cases of hair cell (HC) toxicity, high dosages of aminoglycosides, such as gentamicin and neomycin, affect the hearing threshold [31]. According to the results of our study, CP treatment resulted in a considerable loss of presynaptic ribbons, which was particularly apparent at the base, middle, and apical turns of the cochlea. This was a distinguishing feature of the CP-treated group in comparison to the control group (Figure 3A,B). The IHC’s functional ribbon synapses are the target of CP action. These synapses allow SG neurons to receive the initial auditory signal produced by the IHCs. The auditory transmission to SG neurons and central auditory pathways would be altered if the amount and/or function of these synapses were to change. Therefore, the overall elevations in ABR thresholds were reported in rats treated with CP and this would be caused by the losses in ribbon synapses, or synaptopathy in the cochleae [32]. However, AVN-C pretreatment preserved more presynaptic ribbons in all regions of the cochlea than that in the CP-treated group, while excluding a similar number of presynaptic ribbons as in the control group, which received only the AVN-C carrier. This indicates a potential protective benefit of AVN-C against CP-induced hearing loss (Figure 3A,B). In a recent report, we showed that high-dose methotrexate administration in mice diminished presynaptic ribbons and that this was reversed by AVN-C pretreatment [11]. Additional studies showed that pretreatment with epigallocatechin-3-gallate (EGCG) rescued the decrease in CtBP2 immunoreactivity that occurred after CP treatment in rats [33]. Here, we demonstrate how AVN-C could prevent the loss of presynaptic ribbons caused by CP. The concentration of AVN-C used during in vivo investigations was taken from our earlier reports because it demonstrated NIHL and DIHL protection [10].

CP reduced the viability of HEI-OC1 cells and caused morphological alterations that are indicative of cell death; however, no tested dose of AVN-C (from low to high dose) exhibited any detrimental effects on HEI-OC1 cell survival (Figure 4A,B). The AVN-C concentration for in vitro studies was chosen based on the results of the cell viability assessment on HEI-OC1 cells (Appendix A), and we selected a starting significant low dose of AVN-C, which was 3 μM and it proved to substantially maintain cell viability significantly when it was combined with 30 μM of CP as opposed to CP alone during our experiments (Figure 4C). CP increased the level of ROS in the cells, as determined by the DCF fluorescent probe. The ROS generation by CP was inhibited by pretreatment with AVN-C (Figure 5A,B). CP enhanced ROS production by activating the NOX3 isoform of the NADPH oxidase enzyme system in the organ of Corti HC cultures (UB/OC1 cells), and siRNA against NOX3 prevented this high ROS production [34]. Our studies revealed that the antioxidant activity of AVN-C significantly reduced ROS generation during CP-induced cytotoxicity.

Furthermore, CP elevated levels of inflammatory cytokines (*IL6*, *IL-1β*, *TNFα*, *iNOS*, and *COX2*; Figure 5C), which indicates that the inner ear organs were inflamed and resulted in tissue damage and hearing loss. However, the use of AVN-C to prevent CP-induced ototoxicity markedly reduced inflammation in all examined inflammatory cytokines when administered prior to CP treatment (Figure 5C). CP induces the production of proinflammatory cytokines that are involved in mediating ototoxicity [35]. Etanercept was utilized to reduce CP-induced hearing loss by inhibiting *COX-2*, *iNOS*, and *TNF-α* production in the cultured organ of Corti HCs (UB/OC-1 cells) [36]. Furthermore, In HEI-OC1 cells, CP toxicity induced the release of proinflammatory cytokines, such as *TNF-α*, *IL-1β*, and *IL6*, which cause inflammation. Thus, AVN-C demonstrated its ability to inhibit CP-induced inflammation.

In the cancer cell lines shown in Figure 4D,E, AVN-C had no effect on CP-treated cell survival, showing that AVN-C does not block CP absorption. In addition, Figure 6A,B shows that at 3 h, AVN-C did not stop CP-induced rH2AX foci, proving that it had no influence on CP uptake. Additional evidence suggested that the 12-hour difference in rH2AX between CP and AVN-C supported DNA repair by AVN-C. In all of these investigations, when AVN-C and CP were combined, AVN-C had no influence on CP uptake. Thereby, AVN-C can be utilized to protect against the ototoxic side effects of the drug in cancer patients who are receiving CP. Additional research is ongoing to completely comprehend the mechanism of action of AVN-C.

DNA damage in eukaryotic cells causes a complicated cascade of reactions, such as cell-cycle arrest, re-localization of DNA repair proteins, and, in some circumstances, apoptosis. Within 1–3 min of DNA damage, H2AX, a member of the histone H2A family, undergoes extensive phosphorylation and forms foci at break sites [8]. By phosphorylating histone H2AX, we were able to directly observe the phosphorylated histone molecule at the locations of the DNA breaks in cell nuclei and demonstrated how AVN-C participates in the repair process when CP causes DNA damage. DNA damage was noticed 3 h after the addition of CP to HEI-OC1 cells; within 12 h, the number of immunofluorescent foci increased quickly, at which point there were more foci visible than at 3 h (Figure 6A,B).The results of the cells exposed to AVN-C for 24 h before receiving CP treatment were compared to those of the untreated HEI-OC1 cells (Figure 6A), and this revealed significantly fewer and less brilliant γ-H2AX foci, which demonstrated the potential for AVN-C to contribute to DNA repair (Figure 6A,B). The results of the cells exposed to AVN-C for 24 h before receiving CP treatment were compared to those of the untreated HEI-OC1 cells (Figure 6A), and this revealed significantly fewer and less brilliant γ-H2AX foci, which demonstrated the potential for AVN-C to contribute to DNA repair (Figure 6A,B). Our findings clearly support CP’s role in H2AX foci formation at DNA damage sites and suggested that AVN-C can ensure the change in chromatin structure associated with DSB repair to prevent CP-induced DNA damage and consequent hearing loss (Figure 7).

CP interacts with cochlear tissues, such as OHCs of the organ of Corti, spiral ligament, stria vascularis, and spiral ganglionic cells, to generate a robust ROS response while depleting the antioxidant enzyme system that would scavenge and neutralize the increase in superoxides [37]. The increased ROS production is mainly due to increased lipid peroxidation [38], providing a rationale for using antioxidants such as AVN-C to treat CP ototoxicity. However, further studies are needed to scrutinize the exact mechanism and correlation of AVN-C in preventing CP-induced ototoxicity.

## 4. Materials and Methods

### 4.1. Animal Care

The care and use of animals was undertaken in accordance with the animal welfare guidelines of Chonnam National University. The study protocol was approved and registered by the Committee on the Ethics of Animal Experiments of Chonnam National University (CNUHIACUC-20027). Each group in this study comprised five C57Bl/6 background mice aged 4 weeks. In this study, ketamine (100 mg/kg) and xylazine (10 mg/kg) were used as anesthetics. All surgeries were performed under anesthesia and were designed to minimize suffering. The cochlea was resected for histologic examination. 

### 4.2. Drug Treatment

AVN-C was provided by Professor Jong Hyun Cho, and was synthesized according to the instruction specified in the patent PCT/KR2022/006471. CP was purchased from Sigma Aldrich (Sigma-Aldrich, Inc., St. Louis, MO, USA) in liquid form and was diluted in deionized water. Both drugs were injected intraperitoneally into the mice at the following drug dosages: AVN-C 10 mg/kg one hour before CP 5 mg/kg injected once a day for 4 days; the control group received 5% Kalliphor. The health condition of the mice was good and no mortality was observed. For in vitro investigations with HEI-OC1 cells, AVN-C was dissolved in 0.1% DMSO and administered at 3 µM. CP was given at 30 µM, and the control group received 0.1% DMSO as 1 µL (MTT assay) and 5 µL in other experiments. Under 0.1% DMSO was used in each experiment.

### 4.3. Auditory Brainstem Response Assessment

Prior to the investigation, all animals’ tympanic membranes and external auditory canals were examined. Animals with external or middle ear issues were excluded from the study. We assessed the auditory brainstem response (ABR) from click-to-tone burst stimuli. Mice were sedated with ketamine hydrochloride (80 mg/kg) and xylazine hydrochloride (10 mg/kg). ABR tests were performed in a quiet room at 8, 16, 24, and 32 kHz, as reported previously [39], to the left and right ears at 1 month after drug treatment; body temperature was maintained with the help of a heat-therapy pump (Stryker #TP700, 3800 E. Centre Avenue Portage, Kalamazoo, MI 49002, USA). The TDT’s MF1 Multi-Field Magnetic Speaker was used to optimize the free field utilized to test the hearing range of mice, rats, and guinea pigs [40].

### 4.4. Immunohistochemistry for OHCs and Presynaptic Ribbons

Mice were euthanized with a cocktail of ketamine and xylazine (80 and 10 mg/kg, respectively) before the extraction of the cochleae. Using a 0.5-cc syringe, a hole was created at the apex of each cochlea and the cochlea was then perfused with phosphate-buffered solution (PBS) followed by 4% paraformaldehyde (PFA), then immersed in 4% PFA solution for 1 h with gentle rotation at 4 °C. The cochleae were rinsed twice with PBS before decalcification with 0.12 mM ethylenediaminetetraacetic acid (EDTA) for 1 h on gentle rotation at 4 °C. To reveal the organ of Corti, the bone and stria vascularis surrounding the cochleae were dissected and the tectorial membrane was removed. Three small pieces were cut from each cochlea (apex, middle, and base) and the tissue samples were immersed in blocking buffer for 1 h at room temperature (RT) before being incubated with primary antibodies overnight at 4 °C. After three washing cycles with 0.1% PBS-T (30 min each wash), the samples were incubated in secondary antibodies for 2 h at room temperature. Finally, the samples were washed three times with 0.1% PBS-T for 30 min. Lastly, the samples were stained for 3 min with DAPI and washed in PBS for 30 min. A vector protection solution was used to mount the samples on glass slides and the slides were examined using an LSM 800 laser scanning microscope (Carl Zeiss Microscopy GmbH, Promenade 10, 07745 Jena, Germany). The following antibodies and titers were utilized: C-terminal binding protein 2 (CtBP2; 1:100, # 612044, BD Transduction Laboratories™, 143 NULL, Franklin Lakes, NJ, USA), myosin-7a (1:200, # 25-6791, Proteus, 151 Ramona, CA 92065, USA), Calbindine (1:200, # WC3214704E, Invitrogen, Carlsbad, CA 92008, USA), Prestin (1:1000, # A12379, Cell Signaling Technology, Danvers, MA 01923, USA), and DAPI (1: 10,000, Invitrogen, Carlsbad, CA 92008, USA).

### 4.5. Counting of OHCs and Presynaptic Ribbons

The number of OHCs in the cochlea was ascertained. The cochleae were divided into the apical, middle, and basal turns, and the hair cells in each turn were counted under 200× magnification. The number of hair cells per 100-µm cochlear turn length was averaged for each group (n = 5). 

The lengths of cochlear turns were measured for each study group. Confocal z-stacks of three areas from each cochlea were generated using a high-resolution confocal microscope (LSM 800 laser scanning microscope, Carl Zeiss Microscopy GmbH, Promenade 10, 07745 Jena, Germany). Image stacks were converted using image-editing software. At least 12 IHCs were found in each cochlear turn (apex, middle, and base). We selected three visual areas of 20-µm each for each turn of the cochlea, delineated them with a square, and counted presynaptic ribbons (CtBP2 punctuates) of the IHCs that were identified around IHCs, as well as within nuclei, according to a previously reported method [11].

### 4.6. Cell Culture

The House Ear Institute-Organ of Corti 1 (HEI-OC1) cells were grown under permissive conditions (33 °C) in Dulbecco’s Modified Eagle Medium-high Glucose (Gibco BRL, Gaithersburg, MD, USA), which was supplemented with 10% nonantibiotic fetal bovine serum (Gibco BRL) as previously reported [11]. To elicit cytotoxic effects on HEI-OC1, AVN-C diluted in 0.1% DMSO was added to HEI-OC1 cells at a dose of 3 µM, and CP was diluted in deionized water and administered at 30-µM dose. Cancers cells derived from hypopharyngeal squamous cell carcinoma (SNU-1041) and pharyngeal carcinoma (FaDu) were used in this study to measure the role of the effect of AVN-C or the lack thereof in cisplatin-induced cancer cell death progression. The SNU-1041 cell line was maintained in RPMI 1640 media containing 10% fetal bovine serum (FBS; GIBCO, Grand Island, NY, USA), 100 U/mL penicillin, and 100 g/mL streptomycin (Invitrogen, Carlsbad, CA, USA). Eagle’s modified essential medium (MEM), which contains 10% FBS, 100 U/mL penicillin, and 100 g streptomycin (Invitrogen and GIBCO), from the American Type Culture Collection (ATCC) was used to maintain the FaDu cell line. All cell lines were incubated at 37 °C with 5% CO_2_, which is the standard culture condition.

### 4.7. Cell Viability Assessment

These distinct cell types were subjected to MTT assay after the completion of each treatment regimen. The cells were collected in sterile Eppendorf tubes and the MTT assay was performed 24 h after all treatments in accordance with the manufacturer’s protocol. Absorbance was measured using a Spectra Max 5 Plate Reader and Soft Max Pro 5.2 (Molecular Devices, Sunnyvale, CA, USA); the average OD was used to evaluate viability and the OD of control cells was considered as 100% in HEI-OC1 cells and fold change in cancer cell lines.

### 4.8. Measurement of ROS

The ROS levels within HEI-OC cells were determined using the Reactive Oxygen Species Detection Assay Kit. After 30-min incubation at 37 °C with 5% CO_2_, the cells were suspended. The cell-permeable reagent 2,7-dichlorodihydrofluorescein diacetate (DCFDA) was used to quantify ROS within the cells. HEI-OC cells were washed once in 1× PBS buffer and then suspended in a microplate reader; fluorescence was measured using a flow cytometer at maximal excitation and emission wavelengths of 495 and 529 nm, respectively (BD FACSCalibur^TM^, BD Biosciences, San Jose, CA, USA). Using Kaluza Analysis Software (Beckman Coulter, Inc., Brea, CA, USA), the change in the ROS levels was expressed as a percentage of the control after background subtraction.

### 4.9. H2AX Foci Staining in the Nucleus

The HEI-OC1 cells were grown in 8-well dishes overnight at a ratio of 1000 cells/mL. Next, 3 µmol AVN-C was applied to cells and the cells were incubated for 24 h; this was followed by incubation with 30 µmol CP for 3 and 12 h, respectively. After all treatments were completed, the cells were washed twice with prewarmed PBS before being fixed with 4% paraformaldehyde for 1 h at 4 °C, permeabilized with 0.2% Triton X-100 in PBS for 10 min, and rinsed twice with 0.2% Triton X-100 in PBS. The cells were blocked for 1 h at 4 °C with 2% normal goat serum (NGS) in 0.2% Triton X-100 in PBS, followed by an overnight incubation with the following primary antibodies at a ratio of 1:1000: mouse monoclonal H2AX (Ser-139; #3313712, Millipore Corp, 29 Center St, Burlington, MA 01803, USA). Secondary antibodies against mice that were labeled with Alexa 594 (#148485, Jackson ImmunoResearch, West Grove, PA, USA) were added, and the slides were incubated at room temperature for 1 h. ProLong Gold Antifade reagent containing DAPI was used to mount the slides (Molecular Probes, Eugene, OR, USA). An LSM 800 laser scanning microscope was used to capture images at room temperature (Carl Zeiss Microscopy GmbH, Promenade 10, 07745 Jena, Germany).

### 4.10. RNA Isolation and Real-Time PCR

TRIzol reagent (Invitrogen) was used to extract total RNA from treated HEI-OC1 cells. A full-spectrum spectrophotometer (NanoDrop ND-1000, Technologies Inc., Wilmington, DE, USA) with absorbance measured at A260/A280 nm was employed to determine the amount of RNA. First strand complementary DNA (cDNA) was obtained with an RNA reverse-transcription cDNA synthesis kit (1st strand cDNA Synthesis kit; Takara PrimeScript^TM^, Kusatsu Shiga, Japan) with the following cycle conditions: denaturation at 95 °C for 10 min and 10 s, followed by annealing at 62 °C for 20 s followed by 72 °C for 30 s for 60 cycles; the Taq Master Mix was used for RT-PCR (Bioscience, Dümmer, Germany). The 2^-Ct^ technique was used to determine the expression levels, and the relative mRNA expression was adjusted to that of the glyceraldehyde 3-phosphate dehydrogenase expression (GAPDH) and the experiments were carried out three times. The primers used for each gene are listed in below Table 1:

### 4.11. Statistical Analysis

The Student’s *t*-test or one-way ANOVA with the post hoc Tukey–Kramer comparison test were used for statistical analyses. For all statistical studies, GraphPad Prism version 8.0 was used. *p*-value less than or equal to 0.05 was considered indicative of statistically significant results. The number of repetitions used in each experiment are described in the figure legends.

## 5. Conclusions

In this study, we found that AVN-C alleviated CP-induced oxidative stress and DNA damage in vivo in a mouse model as well as in vitro in HEI-OC1, which indicates a mechanism of shielding cells from CP-induced ototoxicity. Thus, AVN-C is a promising candidate molecule for future evaluation in the treatment of sensorineural hearing loss.

## Figures and Tables

**Figure 1 ijms-24-02947-f001:**
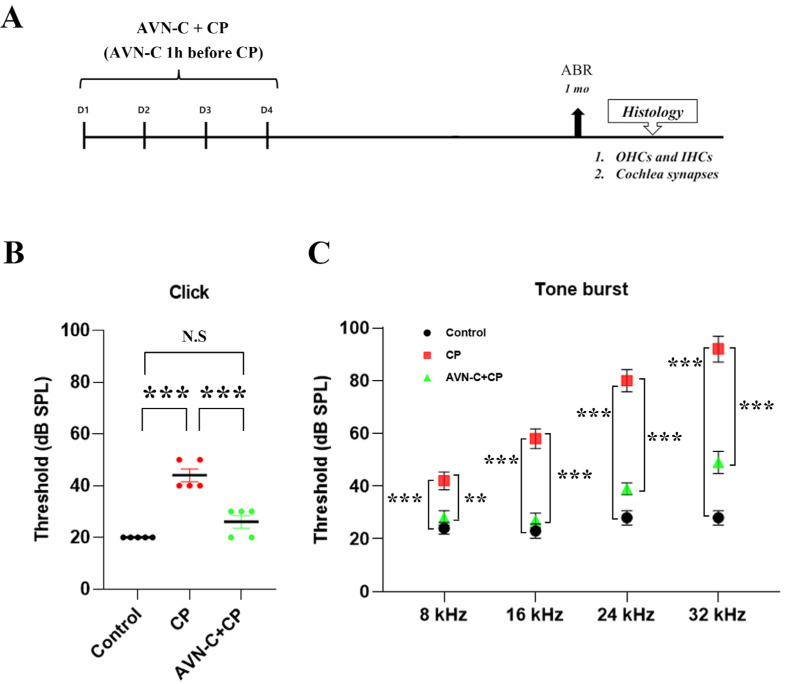
In case of cisplatin-induced ototoxicity, AVN-C improves the auditory brainstem response (ABR). (**A**) For 4 days, AVN-C pretreatment was administered 1 h before cisplatin therapy every day. ABR was performed at 1, 2, and 4 weeks; (**B**) Cisplatin impaired hearing in wild-type mice by causing the threshold shift to increase whereas AVN-C maintained the threshold shift within the normal range (*** *p* ≤ 0.001). (**C**) Cisplatin-induced elevation of thresholds in wild-type mice at high frequencies (16, 24, and 32 kHz) whereas AVN-C mimicked similar results at 8 and 16 kHz and manifested decreased threshold shifts at 24 and 32 kHz (** *p* ≤ 0.01 at 8 kHz, *** *p* ≤ 0.001 at 16, 24, and 32 kHz; AVN-C vs. cisplatin, n = 5). (N.S = not significant).

**Figure 2 ijms-24-02947-f002:**
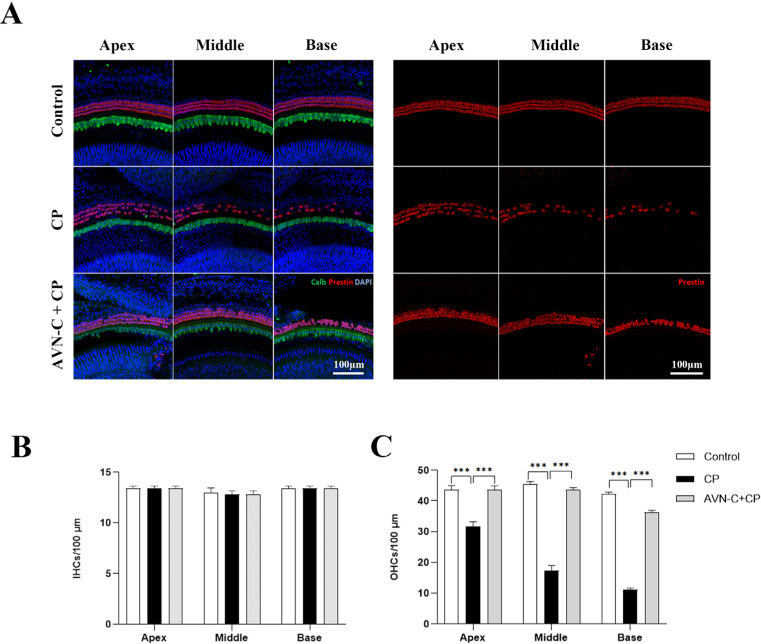
Immunohistochemical staining of the organ of Corti. (**A**) Right panel: images of OHCs stained with Prestin (red); left panel: combined image of Calbindin (green) for IHCs, Prestin for OHCs, and DAPI (blue) staining. (**B**) The IHCs remained unchanged in all three groups (n = 5 each). (**C**) The OHCs were damaged severely by CP treatment (32 ± 3.4, 17 ± 3.4, and 11 ± 1.4 OHCs per 100 µm at apex, middle, and base turns, respectively) whereas AVN-C protected OHCs from the CP-induced loss (44 ± 1.5, 44 ± 1.7, and 36 ± 2.9 OHCs per 100 µm at the apex, middle, and base turns, respectively). The OHCs in the control group were found at the apex (45 ± 3.0), middle, (44 ± 2.1), and base turns (42 ± 1.5 OHCs) per 100 µm, respectively. (*** *p* ≤ 0.001, *p* ≤ 0.05 was considered significant in all experimental groups).

**Figure 3 ijms-24-02947-f003:**
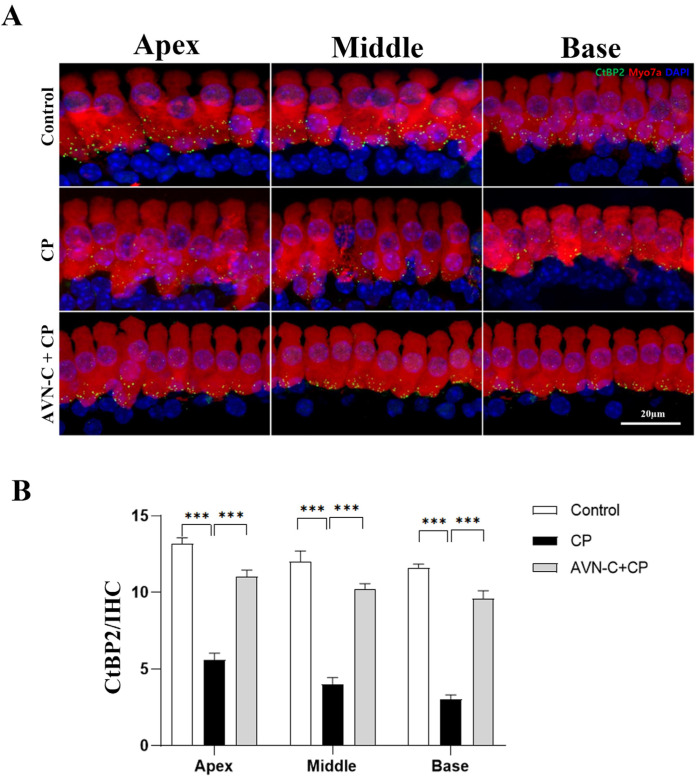
A comparison of presynaptic ribbons based on different treatment plans. (**A**) Representative images of C-terminal binding protein 2 (CtBP2) stained in the IHC. (**B**) Compared to the control and AVN-C-treated groups, the CtBP2 signal is reduced in the CP group, and this indicates the death of presynaptic ribbons (one-way ANOVA, *** *p* ≤ 0.001, at apex, middle, and base turns, Control vs. CP; AVN-C vs. CP). The IHCs were stained with Myosin7a. (n = 5 each group). One-way ANOVA was used, with *p* ≤ 0.05 indicating significance.

**Figure 4 ijms-24-02947-f004:**
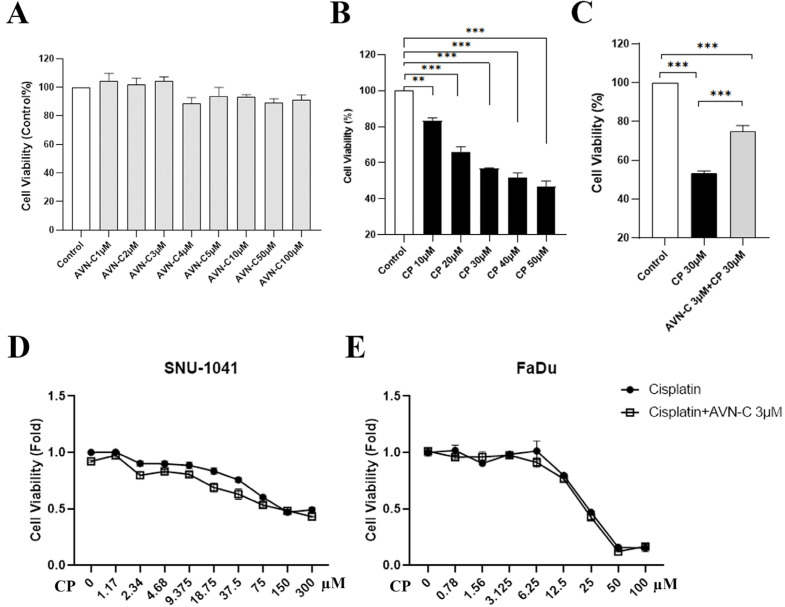
Cell viability assessment by MTT assay. (**A**) AVN-C does not cause toxicity in HEI-OC1 cells. (**B**) Increased concentration of CP resulted in cell death of HEI-OC1 cells (** *p* ≤ 0.01 Control vs. CP 10 µM, *** *p* ≤ 0.001 Control vs. CP 20, 30, 40, 50 µM; n = 3 each). (**C**) AVN-C maintained cell survival in CP-induced ototoxicity (*** *p* ≤ 0.001, AVN-C vs. CP; n = 3 each). (**D**,**E**) AVN-C did not influence the effect of CP on the survival of cancer cell lines (n = 3 each). One way-ANOVA was used and *p* ≤ 0.05 indicated significance.

**Figure 5 ijms-24-02947-f005:**
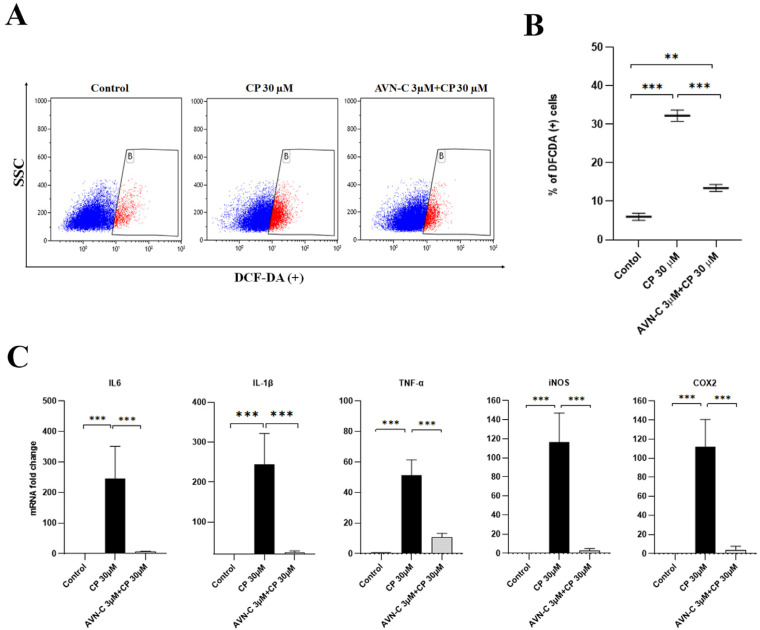
AVN-C reduces apoptotic bodies in CP-induced ototoxicity in HEI-OC1 cells. (**A**,**B**) ROS significantly increased when 30 µm CP was administered to HEI-OC1 cells (*** *p* ≤ 0.001, Control vs. CP and AVN-C vs. CP; ** *p* ≤ 0.01, Control vs. AVN-C+CP ) and AVN-C mitigated CP cytotoxicity by protecting the HEI-OC1 cells against CP-induced damage (*** *p* ≤ 0.001; n = 3 each). Demarcated (**B**) area in red shows DCFDA positive. (**C**) The levels of inflammatory cytokines (*IL6*, *IL-1β*, *TNF-α*, *iNOS*, and *COX2*) that are known to trigger inflammation of auditory hair cell death were all upregulated in the CP-treated group (*** *p* ≤ 0.001, Control vs. CP), and AVN-C significantly downregulated the expression of these genes (*** *p* ≤ 0.001, AVN-C vs. CP; n = 3 each). One way-ANOVA was used and *p* ≤ 0.05 indicated significance.

**Figure 6 ijms-24-02947-f006:**
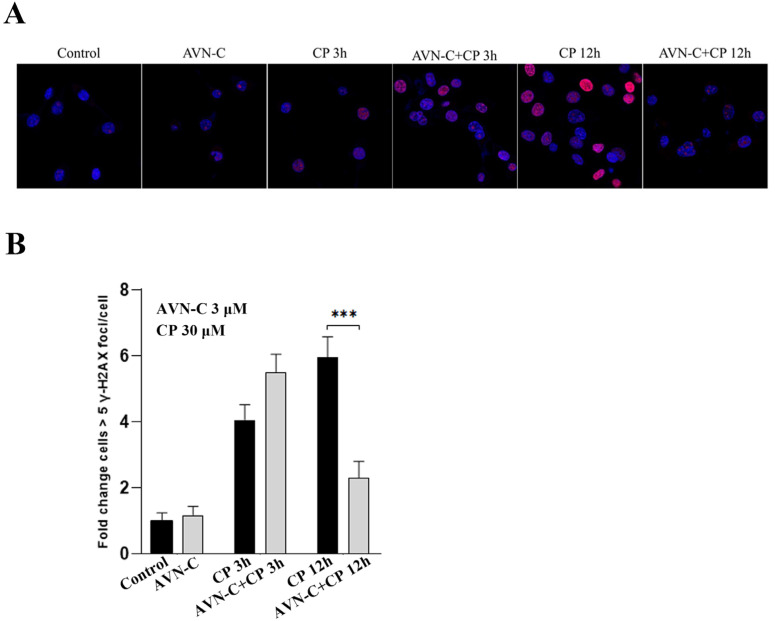
AVN-C prevents DNA damage caused by CP. AVN-C 3 µM and CP 30 µM were used in this study. (**A**) Representative images of γH2AX-stained HEI-OC1 cell. (**B**) γH2AX levels in relation to exposure to CP and AVN-C at different timepoints was as follows: Control (1 ± 0.2), AVN-C (1.2 ± 0.3), CP 3 h (4.1 ± 0.5), AVN-C+CP 3 h (5.5 ± 0.6), CP 12 h (6.1 ± 0.6), and AVN-C+CP (2.3 ± 0.5) fold-change cells > 5 γH2AX foci/cell and (*** *p* ≤ 0.001, CP vs AVN-C. + CP-treated groups at 12 h). One-way ANOVA was used and *p* ≤ 0.05 was considered significant. Image magnification 40X was used.

**Figure 7 ijms-24-02947-f007:**
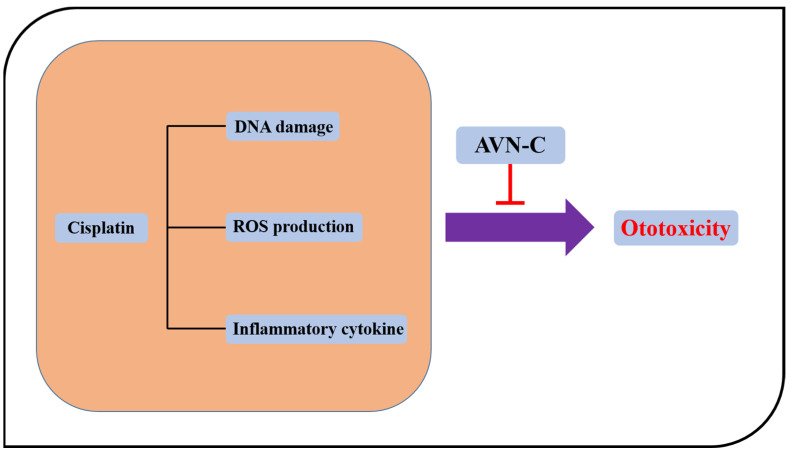
Proposed mechanism of action by AVN-C. CP damages DNA and induces the production of ROS and inflammation whereas AVN-C inhibits both ROS and inflammation and DNA-involved damage. Protects against is shown by the red icon below AVN-C.

**Table 1 ijms-24-02947-t001:** List of primers.

iNOS forward	5′-GCATGGAACAGTATAAGGCAAACA-3′
iNOS reverse	5′-GTTTCTGGTCGATGTCATGAGCAA-3′
COX2 forward	5′-GCATGGAACAGTATAAGGCAAACA-3′
COX2 reverse	5′-GTTTCTGGTCGATGTCATGAGCAA-3′
IL-1β forward	5′-GCTGCTTCCAAACCTTTGAC-3′
IL-1β reverse	5′-AGGCCACAGGTATTTTGTCG-3′
TNFα forward	5′-CCACCACGCTCTTCTGTCTA-3′
TNFα reverse	5′-CACTTGGTGGTTTGCTACGA-3′
IL-6 forward	5′-TCCAGTTGCCTTCTTGGGAC-3′
IL-6 reverse	5′-GTACTCCAGAAGACCAGAGG-3′
GAPDH forward	5′-ACCACAGTCCATGCCATCAC-3′
GAPDH reverse	5′-TCC ACC ACC CTG TTG CTG TA-3′

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
