# Peer review of "Protective Effect of Avenanthramide-C on Auditory Hair Cells against Oxidative Stress, Inflammatory Cytokines, and DNA Damage in Cisplatin-Induced Ototoxicity"

_ijms, 2023, doi:10.3390/ijms24032947_

Round 1

Reviewer 1 Report

The authors demonstrated that AVN-C had a strong protective effect against cisplatin-induced ototoxicity through the inhibition of ROS and DNA damage. However, the methods should be improved to clearly prove the results. There are a lot of questions that need to be addressed.

  1. For the prevention or treatment of anti-ototoxicity agents, the dosing interval between cisplatin and anti-ototoxicity agents is crucial for efficiency.

In the method part, the authors didn't mention the dosing interval.

  1. Because AVN-C was administrated by IP, the concentration of AVN-C in perilymph should be detected. Safety evaluation should be designed to estimate if there were side-effects after  AVN-C administration by IP.
  2. It is better to exhibit the counts of OHC and synapticapatic ribbon of the whole organ of coti.
  1. In figure 1 and figure 3, the figure legend showed "(***p>=0.001)." and "(**p>=0.01 at 8 kHz,", which may be incorrect.

Reviewer 2 Report

This manuscript by Umugire et al describes experiments which explore the protective effect of Avenanthramide-C (AVN-C) towards cisplatin-induced ototoxicity. Several points require clarification:

1. There are several places where the text is difficult to follow, e.g. the sentences beginning on lines 20 and 27 do not scan; the paragraph beginning on line 46 appears with no context within the Introduction and the terms rH2AX and DSBS are not defined. On line 59, "wipe out" is not a scientific term.

2.  In the Results section, several instances show values in the text that are unnecessary and could be removed since they are present in the figure legends: Section 2.2/Figure 2; Section 2.7/Fig 6B. Also please clarify the discrepancy between the value for the 12hr time point for cisplatin in the text of 6.1 and in the Fig 6B legend of 6.0.

3.  In Figure 1A, the schematic indicates that ABR data were collected at 1 week, 2 weeks and 1 month post cisplatin treatment for all groups. The data shown are presumably from a single time point, perhaps 1 month as stated in the Discussion (line 213)? The time point should be stated in the figure legend and text and the 1 and 2 week data should also be included.

4.  In Figure 2A, what do the two sets of panels represent? The figure legend only states that these are OHCs, but clearly IHCs are stained in the left hand panel. Also, please state in all of the figures what stain the colors represent. The IHCs in the Control Middle panel of the lefthand side of Figure 2A appear to be less dense from the image shown, although this is not reflected in the quantification in Figure 2B. Please explain.

4.  Section 2.4 referring to Figure 4A and B, needs to be clarified since on line 138 it is not the levels of cisplatin and AVN-C that were evaluated. Presumably the authors mean the concentrations?

5.  In Figure 6, there is no mention of the concentrations of cisplatin and AVN-C that were used.

6.  In general, the Discussion is poorly organized and largely consists of a restatement of the Results, e.g. lines 247 through 283. This should be redone with an emphasis on how the current data relate to other studies in the field.

7.  The authors provide a simplistic mechanistic hypothesis based on their findings (Figure 7), but do not discuss other possibilities to explain their data, which cannot currently be ruled out. Since AVN-C was either given as a pretreatment prior to cisplatin or with co-treatment, the possibility that AVN-C binds to and neutralizes cisplatin should be considered (the only approved drug for cisplatin-induced hearing loss, Pedmark, works through this mechanism). Additionally, can the authors rule out that the possibility that AVN-C inhibits the uptake of cisplatin by cells? In the in vivo experiments, can the authors rule out the possibility that AVN-C alters the pharmacokinetics of cisplatin to reduce its concentration in the inner ear? If there are no data to rule out these alternative mechanisms, they should be acknowledged by the authors as possibilities in the Discussion.

8.  In the Methods it is stated that 5% Kalliphor was used in the Control group and presumably this was the vehicle used for AVN-C. If so, that should be stated in the Methods.

9.  Where statistics are shown in the Figures, details should be included in the figure legends of the statistical test that was used for that set of data, ie one-way ANOVA also including F-values, t-test etc.

10. On line 354 it is stated that AVN-C was diluted in DMSO for in vitro experiments. DMSO itself has been shown to bind to cisplatin and reduce its biological activity (Hall et al, 2014, Cancer Res. 24:3913) or with regards to ototoxicity can in certain circumstances potentiate cisplatin effects (Uribe et al, 2013, PLoS One 8: e55359). Therefore, it is critical to control the amount of DMSO used in both AVN-C and control situations to ensure that the effects observed are truly due to AVN-C and not to varying amounts of DMSO. For all in vitro experiments, the authors should state the amount of DMSO that was included in both control and AVN-C conditions.

11.  The authors used CtBP2 staining as a marker for presynaptic ribbons. At several places, they describe this as "synapses", e.g. Figure 3 legend, paragraph starting line 232. They should be careful to describe this only as presynaptic ribbons because to make a definitive assessment of IHC synapses a co-staining with a post-synaptic component is required , e.g. GluA2. This is the standard in the field.

Reviewer 3 Report

The file with the reviewer comments is attached.

Round 2

Reviewer 1 Report

Agree

Author Response

The authors appreciate your valuable time to review their manuscript. 

Reviewer 2 Report

The authors have responded to the majority of my comments, but some items require additional clarification:

Q5: the authors added the concentrations used in the Fig 6 legend, but these are expressed as umol (an amount) and should be uM (a concentration), please correct.

Q7: the authors addressed some of the alternative explanations for mechanism, but not the possibility that AVN-C changes the cisplatin pharmacokinetics. If they have no data to rule this out, they should acknowledge this possibility in the Discussion. The reference added to support penetration of AVN-C into the perilymph raises some additional related concerns (ref 10). In that paper, the authors report perilymph levels of approx 1ng/ml or less after 10mg/kg AVN-C ip in mice. This equates to a concentration of 3nM AVN-C or less (MW = 315) and consequently is 3 orders of magnitude less than the concentrations required for effects on cisplatin in the in vitro experiments (3uM). This is a considerable discrepancy which should be acknowledged by the authors and further supports the need to include other potential explanations for the in vivo effects as noted above.

Q10: it is still not clearly stated what DMSO concentration was present in the in vitro assays. In the methods the authors now state: "AVN-C was dissolved in 0.1% DMSO and administered at 3 µM. CP was given at 30 µM, and the control group received 0.1% DMSO as 1 µL (MTT assay) and 5 µL in other experiments." Please simply state the final DMSO concentration in each assay (not in the solution added to the assay), and please confirm that the final DMSO concentration was the same for all conditions compared (control, cisplatin alone and cisplatin plus AVN-C). If this was not the case, and DMSO concentrations differed between compared groups, this should be stated and acknowledged as an uncontrolled variable that may have influenced the results.
